# The Antioxidant and Anti-Aging Effects of Acetylated Mycelia Polysaccharides from *Pleurotus djamor*

**DOI:** 10.3390/molecules24152698

**Published:** 2019-07-24

**Authors:** Huaping Li, Huajie Zhao, Zheng Gao, Xinling Song, Wenshuai Wang, Fangfang Yuan, Yanbo Feng, Yiwen Zhang, Jianjun Zhang, Shuliang Zhang, Le Jia

**Affiliations:** 1Dezhou Academy of Agricultural Sciences, Dezhou 253015, China; 2College of Life Science, Shandong Agricultural University, Taian 271018, China

**Keywords:** acetylated mycelia polysaccharides, *Pleurotus djamor*, antioxidant effects, anti-aging effects

## Abstract

The present work mainly describes the preparation of acetylated mycelia polysaccharides (AMPS) from *Pleurotus djamor* and investigates the antioxidant and anti-aging effects in d-galactose-induced aging mice. The optimized procedure indicates the acetyl substitution degree of AMPS is 0.54 ± 0.04 under the conditions of a reaction time of 56 h, a reaction temperature of 37 °C, and 4 mL of added acetic anhydride. The in vitro analysis and in vivo animal experiments indicate that the AMPS could alleviate the aging properties by scavenging the radicals, elevating the enzyme activities, and reducing the lipid contents. As for serum levels, the AMPS can improve the serum biochemical indices and enhance immunological activity. The histopathological observations indicate that the injuries to the liver, kidney, and brain can be remitted by AMPS intervention. The characterization showed that AMPS was one kind of β-pyranose with the weight-average molecular weights of 3.61 × 10^5^ Da and the major monosaccharides of mannose and glucose. The results suggest that AMPS can be used as a dietary supplement and functional food for the prevention of aging and age-related diseases.

## 1. Introduction

It has been reported that aging, as an inevitable and physiological process which can induce the gradual loss of body function and increase the probability of age-related diseases, including diabetes, cognition impairment, cancer, liver damage, Parkinson’s disease, and atherosclerosis, has become a considerable challenge in health and life ability [1,2,3]. d-galactose (d-gal) can react with free amines of amino acids as a physiological reducing sugar and further form mashing terminal products. Mammals that are exposed long-term to excess d-gal can fall victim to induced accelerated aging, including regression of the hematopoietic system and bone marrow [4]. Oxidation is essential for many organisms in all biological processes related to energy metabolism. However, excess free radicals can be scavenged by antioxidants, which interact with free radicals against oxidative stress [5]. Oxidative stress can be generated by producing reactive oxygen species (ROS) in the application process of d-gal. 

However, the long-term use of chemically synthesized antioxidants can be toxic and carcinogenic, so it is important to find effective alternative natural antioxidants. Edible and medicinal mushrooms are rich in polysaccharides, balanced nutrients, dietary fiber, and other compounds which are physiologically beneficial to humans [6]. It was proven that polysaccharides from mushrooms play significant roles in refraining from oxidative damage in living organisms as inartificial free radical scavengers and have better effects than synthetic antioxidants [7]. Published studies have demonstrated that polysaccharides are abundant and can be isolated, purified, and structurally characterized and possess biological functions, such as anti-tumor, hepatoprotective, anti-inflammatory, anti-viral, antioxidative, and anti-microbial activities [8,9]. *Pleurotus djamor*, which is classified in the genus *Pleurotus* of the family Pleurotaceae, has been authenticated as an edible and medicinal mushroom and verified to have disease-resistance and health protective effects. Moreover, polysaccharides from *P. djamor* have been affirmed to possess multiple biological activities, including antioxidant, antitumor, and immune adjustment activities in many certified research examples [10]. Additionally, the purification, characterization, and hepatoprotective activities of mycelia zinc polysaccharides by *P. djamor* were studied by Zhang et al. [11], but acetylation modification of *P. djamor* has not been researched before. Acetylation is an important modification method for polysaccharides’ branched chains and the activity of polysaccharides is closely related to the structure, molecular weight, and solubility of polysaccharides [12]. An acetyl group can change the extension of polysaccharides, lead to the exposure of a polysaccharides hydroxyl group, increase the solubility of polysaccharides in water, improve the physiological activity of polysaccharides, and make polysaccharides be used effectively [13]. Furthermore, previous literature found that acetylated modification polysaccharides can significantly improve their cytoprotective and antioxidant activities, including resisting reactive oxygen [14].

From the above, polysaccharides can serve as age-defying, momentous, natural edibles that can assist chemical medicines. However, there are few studies that pay close attention to the anti-aging function of the acetylated mycelia polysaccharides (AMPS) of *P. djamor*, so it is necessary to explore this. The present experiment was aimed to describe the antioxidant activity in vitro, anti-aging activities in vivo, and evaluate the effects of AMPS on d-gal induced-aging mice.

## 2. Results

### 2.1. AMPS Acetyl Substitution Degree

Results of single-factor and response surface optimization experiment are shown in Figure 1. The predicted value of substitution degree (DS) can be calculated by the following formula: DS = 0.54 + 0.02 × A − 7.5×10^−3^ × B − 0.015 × C + 0.01 × A × B + 0.01 × A × C − 0.015 × B × C − 0.025 × A^2^ − 0.085 × B^2^ − 0.03 × C^2^. The optimal conditions for acetyl substitution were determined as a reaction time of 56 h, a reaction temperature of 37 °C, and 4 mL of added acetic anhydride (Appendix A). 

The absorbance at 500 nm was 0.38 ± 0.03 and the concentration of solution in the 50 mL volumetric flask was 0.025 ± 0.003 mg/mL with the acetyl standard curve (Figure 2). The w2 value was 12.50 ± 0.18 according to the formula and the DS value was 0.54 ± 0.04.

### 2.2. FT-IR, HPLC, and NMR Analysis

The FT-IR spectrum of AMPS is presented in Figure 3A. A strong and broad absorption peak at 3700–3100 cm^−1^ for –OH vibrations, two peaks for C–H within the range of 3000–2800 cm^−1^, and several compact absorption peaks in the region of 1200–1000 cm^−1^ for C–OH and the C–O–C glycosidic band vibrations existed for AMPS. All these characteristic absorptions indicated that the FT-IR spectrum belonged to the acetylated mycelia polysaccharides from *P. djamor*. Additionally, the figure showed a doughty absorption between 1655 cm^−1^ and 1550 cm^−1^ (1577.37 cm^−1^, accurately) and was typical of N-acetylglucosamine residues. We ascertained the successful acetylation and the remarkable substitution effect of AMPS through this extremely obvious absorption peak [15,16]. There was also a symmetrical, strong, stretching vibration carboxyl peak at 1410.39 cm^−1^. The absorption peaks at 1000–700 cm^−1^ were characteristic peaks formed by α-pyranose and β-pyranose and the absorption peaks near the 890 cm^−1^ band (880.17 cm^−1^ in the FT-IR spectrum) were characteristic peaks formed by the –CH bending vibration of β-pyranose, meaning that the AMPS was a kind of β-pyranose [17].

The HPLC chromatogram indicated that AMPS was a heterogeneous polysaccharide. The Mw (weight-average molecular weight), Mn (number-average molecular weight), and Mz (mean grain size) of the AMPS were 3.61 × 10^5^ Da, 2.06 × 10^5^ Da, and 6.85 × 10^5^ Da, respectively. The ratio value of Mz/Mw was 1.90.

The ^1^H-NMR spectrum of AMPS (Figure 3B) represented the peak of D_2_O at 4.69 ppm; two fairly strong signals of anomeric protons at 4.89 and 5.30 ppm were observed, possibly indicating that the AMPS was mainly composed of two types of monosaccharides. The ^1^H-NMR spectrum also showed the set of wide and intense signals between 3.0 and 4.0 ppm, which manifested in CH_2_–O and CH–O groups present in AMPS. Moreover, the anomeric regions of 0.9–2.0 ppm and 3.2–5.5 ppm were the general representative carbohydrate peaks. The signal of D_2_O in the ^13^C-NMR spectrum (Figure 3C) was at 181.49 ppm and the AMPS potentially contained at least two monosaccharides due to the anomeric carbon signals at 97.64 and 97.82 ppm, which agreed with the ^1^H-NMR spectrum analysis. Other important signals in the present spectra were those in the area of 60–80 ppm, where they were related to C_2_, C_3_, C_4_, C_5_, and C_6_ from the glycosidic ring. In addition, the absence of α-anomeric configurations in AMPS could be reflected because there were no signals during 98–102 ppm. The wide and intense peaks at 60–110 ppm suggested the presence of β-(1→3) glucans in AMPS [11,18].

Comparing the retention time of the standards (Figure 3D), the AMPS was composed of mannose, ribose, rhamnose, glucuronic acid, glucose, galactose, xylose, and fucose in a percentage composition of 12.51%, 3.96%, 2.0%, 0.99%, 60.49%, 18.02%, 0.92%, and 1.10% (Figure 3E). Mannose and glucose among the above eight kinds of monosaccharides may be the two types of monosaccharides in the ^1^H-NMR spectrum; the monosaccharide composition of AMPS was ascertained by HPLC.

### 2.3. Antioxidant Properties In Vitro

In this experiment, to determine the antioxidant activities of the AMPS in vitro, reducing power analysis, scavenging 1,1-diphenyl-2-picrylhydrazyl (DPPH) radical assay, and scavenging assays on hydroxyl and superoxide radicals were conducted and the results are presented in Figure 4. The reducing power of AMPS was increased with the increase of sample concentrations (Figure 4A). At a concentration of 1000 mg/L, the absorbance at 700 nm reached 0.47 ± 0.02 and the 50% effective concentration (EC_50_) level was 620 mg/L. It can be seen from Figure 4B that the scavenging DPPH radical role increased with the increasing concentrations of the polysaccharides. At a concentration of 1000 mg/L, the scavenging DPPH radical effect was 64.09 ± 1.34% and the 50% inhibitory concentration (IC_50_) level was 580 mg/L. The scavenging hydroxyl radical abilities with concentrations ranging from 100 to 1000 mg/L were depicted in Figure 4C. The results indicated that AMPS had a potential scavenging result with a concentration-dependent manner. The scavenging hydroxyl radical value reached 28.91% ± 1.24% under a concentration of 1000 mg/L and the IC_50_ level was 715 mg/L. The dose-response curve for scavenging superoxide radicals activities of AMPS are shown in Figure 4D. The scavenging rate reached 55.05% ± 1.96% at 1000 mg/L and the IC_50_ level was 585 mg/L.

### 2.4. Oral Chronic Toxicity Studies

Obviously, no significant changes of behavioral, autonomic, or toxic responses, as well as no deaths, were observed in the mice treated with AMPS, even at doses of 900 mg/kg/d, not only at the end of the experiment, but also during the investigation period.

### 2.5. Effects of AMPS on Body Weight, Liver, and Kidney Indexes

The effects of AMPS on body weight, liver, and kidney indexes in aging mice induced by d-gal are displayed in Table 1. At the onset of the experiments, there was no significant difference (*p* < 0.05) between the body weights of model control (MC) and the other groups. However, at the end of the experiment, the aging mice in the MC group exhibited a significant loss of weight when compared to the mice in the normal control (NC) group (*p* < 0.05). When compared with the mice in the MC group, the mice in the two dose groups exhibited significant improvement in the loss of body weight after the administration of the polysaccharides. Additionally, the results indicated that high dosage group (H-AMPS) showed a stronger effect on improving body weight, which was very close to the values in the positive control (PC) group. After administering AMPS at a high dose of 400 mg/kg/d, the weights increased by 18.31% ± 0.09% compared with the MC group (*p* < 0.05) and the weights at a low dose of 200 mg/kg/d increased by 12.62% ± 0.08% (*p* < 0.05).

Furthermore, the liver, kidney, and brain indexes in the mice were also investigated and are expressed as liver, kidney, or brain weight/body weight (g/100 g) in Table 1. There were significant augmentations in the liver index and kidney index that could be observed in the aging mice of the MC group when compared to the NC group (*p* < 0.05), indicating that liver and kidney damage occurred after six weeks post-d-gal injection. Moreover, the liver and kidney indexes were found to decrease after the treatment with AMPS at two dosages. In the H-AMPS group, the liver and kidney indexes were severely reduced by 33.03% ± 0.16% and 32.86% ± 0.56%, which were higher than the changes observed in low dosage group (l-AMPS) (31.17% ± 0.05% and 21.33% ± 0.33, respectively), indicating that both dosages of AMPS had certain protective effects against the d-gal-induced liver and kidney injury. However, there were no significant differences (*p* > 0.05) in brain index. A possible reason for this is that the quality of the brain was relatively light and therefore could not accurately reflect the brain damage caused by d-gal-induced aging and the alleviating effect of polysaccharides in terms of brain index. Therefore, the subsequent ELISA method was used to detect the nerve growth factor (NGF), tumor necrosis factor-α (TNF-α), and B-lymphoma factor-2 (Bcl-2) in the serum, which largely reflected the influence of aging on brain tissue (Figure 6).

### 2.6. Anti-Aging Properties In Vivo

#### 2.6.1. Effects of AMPS on SOD, GSH-Px, and CAT Activities

When compared to the normal control group, mice in the model control group showed some abnormal behavior characteristics, such as slow reactions, reduced feeding, and sparse hair. The death ratio was also higher, suggesting the success of aging model construction in the behavioral aspects of aging mice. The mice in the dose groups, which were given 400 mg/kg/d and 200 mg/kg/d respectively, showed better manners when compared with those in the MC group. So the reliability of the following experimental samples can be ensured, further studies on the anti-aging properties in vivo should be carried out [19].

The results of the superoxide dismutase (SOD), glutathione peroxidase (GSH-Px), and catalase (CAT) activities in the liver, kidney, and brain are distinctly exhibited in Figure 5. The SOD, GSH-Px, and CAT activities of the mice in the MC group decreased dramatically all in livers, kidneys, and brains when compared with the NC group. After six weeks of intraperitoneal injection administration, the GSH-Px activities in the livers, kidneys, and brains of H-AMPS (200.38 ± 4.37 U/mg prot (U/mg protein) in the liver, 121.45 ± 2.09 U/mg prot in the kidney, and 56.98 ± 1.22 U/mg prot in the brain) increased significantly (*p* < 0.05) compared with those of the model control group (56.12 ± 1.05 U/mg prot in the liver, 42.33 ± 1.89 U/mg prot in the kidney, and 21.19 ± 1.54 U/mg prot in the brain) (Figure 5A–C). The SOD activities of the livers, kidneys, and brains in the H-AMPS group (124.37 ± 2.09 U/mg prot in the liver, 73.55 ± 2.01 U/mg prot in the kidney, and 54.96 ± 0.49 U/mg prot in the brain) increased significantly (*p* < 0.05) compared with those of the model control group (81.54 ± 1.56 U/mg prot in the liver, 52.14 ± 1.23 U/mg prot in the kidney, and 39.35 ± 0.86 U/mg prot in the brain). The administration of AMPS significantly (*p* < 0.05) enhanced SOD activities in a dose-dependent manner, especially in the H-AMPS group. For example, the SOD activities in the brain were increased by 39.67% ± 0.43% compared with the MC group, which was higher than that of the positive control group (35.93% ± 0.17%) (Figure 5D–F). The CAT activities in the liver, kidney, and brain reached 135.32 ± 2.43 U/mg prot, 180.75 ± 4.86 U/mg prot, and 179.18 ± 2.71 U/mg prot, respectively, following treatment with AMPS at a dosage of 400 mg/kg/d. These values were significantly higher than those observed in the MC and increased by 3.13% ± 0.48% in the kidney and 6.36% ± 0.03% in the brain compared with the positive group (Figure 5G–I).

#### 2.6.2. Effects of AMPS on MDA and LPO Contents

A significant increase in malondialdehyde (MDA) and lipid peroxide (LPO) contents (Table 2) was observed in the model control group compared with the normal control group. The MDA contents of H-AMPS and L-AMPS in the liver were reduced by 56.85% ± 0.57% and 29.22% ± 0.24% with dose-dependent patterns at the test concentrations compared with the MC group, respectively. There were also significant decreases in MDA levels observed both in the kidney (61.09% ± 0.57%) and in the brain (33.40% ± 0.16%) after the polysaccharide intragastric administration as compared to the MC group, indicating that AMPS could relieve the aging of mice undergoing d-gal treatment. Similarly, the LPO contents in both dose groups were also lower than those in MC group, especially for the H-AMPS group (42.6% ± 0.46% of liver, 47.78% ± 0.43% of kidney, and 42.47% ± 0.41% of brain). The potential capacity of AMPS to relieve the injury in the liver, kidney, and brain was also presented by values of LPO with a dose-dependent tendency. 

#### 2.6.3. Effects of AMPS on the Serum Biochemical Index

As shown in Figure 6, the mice in the MC group showed serious organic damage, which was evidenced by significant increase in the activities of alanine transaminase (ALT), cholinesterase (CHE), creatine kinase (CK), and aspartate aminotransferase (AST), as well as levels of total bilirubin (TBIL), creatinine (CREA), total cholesterol (CHO), and glucose (GLU) (all with *p* < 0.05) when compared with those in the NC group. The TBIL and CHO levels and CK activities were 1.2 ± 0.1 μmol/L, 2.7 ± 0.2 mmol/L, and 1123 ± 53 U/L in mice treated with AMPS at a dosage of 400 mg/kg/d, which were significant lower than that in the MC group (2.5 ± 0.1 μmol/L, 3.7 ± 0.1 mmol/L, and 2183 ± 113 U/L, respectively, *p* < 0.05) (Figure 6B,G,E). Furthermore, the GLU level in the H-AMPS group decreased by 47.81% ± 0.19% compared with the MC group (15.77 ± 0.63 mmol/L) (Figure 6H). The AST and CHE activities (123 ± 3 U/L and 110 ± 1 U/L, respectively) of mice exerted 400 mg/kg/d polysaccharides were significantly decreased (Figure 6C,F). ALT activities and CREA levels showed the H-AMPS group (42 ± 6 U/L and 63 ± 2 μmol/L, respectively) had superb effects on d-gal-induced aging mice when compared with the MC group (*p* < 0.05) (Figure 6A,D). 

In this experiment, the levels of inflammatory cytokines NGF, TNF-α, and Bcl-2 in the serum were commonly used as biochemical markers for tissue damage. As shown in Figure 7, the levels of NGF and Bcl-2 in the MC group were evidently decreased when compared with those in the NC group (*p* < 0.05), while the value of TNF-α showed an obvious uptrend. The results illustrated that the inflammatory reaction appeared in various kinds of tissues, including liver, kidney, and brain. Interestingly, after the treatment with AMPS at a dosage of 400 mg/kg/d, NGF levels reached 112.121 ± 7.01 pg/mL with an increasing rate of 313.14% ± 0.29% (Figure 7A), TNF-α levels reached 123.362 ± 4.00 ng/L decreased by 40.28% ± 0.56% compared with the MC group (Figure 7B), and Bcl-2 levels reached up to 33.133 ± 0.28 μg/mL with an increasing rate of 13.78% ± 0.07% when compared with the MC group (Figure 7C), respectively. These results indicated that AMPS could regulate the expression of these cytokines commendably.

#### 2.6.4. Histopathological Observations

In the current work, histopathological observations of the liver, kidney, and brain samples were performed to corroborate the evidence from biochemical analyses (Figure 8). As shown in Figure 8A, when compared with the normal hepatic architectures of mice in the NC group with normal cell structures, large circular nuclei located in the center of the cell, rich cytoplasm, and clear cytomembranes, the liver sections of the MC group showed severe hepatocyte apoptosis, hepatocellular swelling, fatty accumulation, a loss of cellular boundaries, and other inflammatory changes. The histopathologic pathological injuries of mice fed with AMPS were ameliorated, according to a diminution of necrotic zones and color change, especially at the dose of 400 mg/kg/d. In a histopathological study of the kidney (Figure 8B), the tissue slice of the NC group exhibited glomeruli that were clear in structure, normal in size and shape, clear in structure and interstitial area of tubules. There was also no dilation of tubular lumen and no hyperplasia of fibrous tissue in the renal interstitium for the NC group, while the MC group showed irregular interstitial areas of the renal tubules, dilated lumens, inflammatory cell infiltration, and fibrous tissue proliferation. The mice in the dose groups of polysaccharides appeared to have less renal injury compared with the MC group. In the study of the effect of AMPS on the morphology of brain necrotic cells (Figure 8C), neurons in the hippocampal cortex of the NC group had intact morphology, with the nuclei and cytoplasm in the center, clear staining, and a large number and good arrangement of neurons. The necrotic cells in the MC group were disordered and the nerve fibers were loose and vacuolated. In particular, the number of vacuolar neurons in the H-AMPS group was significantly lower than that in the MC group, meaning that the degree of degeneration and necrosis of nerve cells in the polysaccharide group was decreased. 

## 3. Discussion

Aging patients are more likely to suffer from chronic diseases, including cancer, hypertension, atherosclerosis, idiopathic pulmonary fibrosis, osteoarthritis, Alzheimer’s disease, Parkinson’s disease, cataracts, and so on. Therefore, the aging mechanism and anti-aging have become more important in research. In this work, the d-gal-induced aging model was adopted. The mechanisms of this model include mitochondrial disorders, gene dysfunction, protein homeostasis imbalance, nutritional metabolic signal changes, stem cell injury, inflammatory aging, and oxidative stress, which was the mechanism studied in this paper [20,21]. Oxidative stress induces the destruction of molecular structures and redox signaling pathways that interfere with the oxidation–antioxidant balance [22]. An appropriate level of oxidative stress is critical for cell survival and homeostasis, but severe oxidative stress can lead to cell damage and death [23]. Edible and medicinal fungal polysaccharides have been studied for their various beneficial effects on the human body [24]. Natural polysaccharides are poly-hydroxyl compounds and nucleophilic substitution can occur on the active hydroxyl group under appropriate conditions, generating corresponding polysaccharide esters [25]. In recent years, many reports have involved the chemical modification of polysaccharides, including acetylation, which can improve the biological activity of polysaccharides through molecular modification [26]. In this experiment, only the relevant content of AMPS at a substitution degree of 0.54 ± 0.04 was carried out and the conditions for obtaining the optimal substitution degree of mycelium polysaccharides from *Pleurotus ostreatus* need to be further studied. It was shown that the weak dissociation energy of the O–H bond caused hydrogen to be donated to a superoxide anion easily and the scavenging radical capacity was related to the amount of the –OH group [27,28]. The content of C=O and C–O bands rose because of the acetylation of mycelium polysaccharides (MPS) from *P. ostreatus* and the antioxidant activity increased with the change of polysaccharide elongation [13]. Data from present work indicated that the FT-IR spectrum of AMPS not only changed between 1655 cm^−1^ and 1550 cm^−1^, which was caused by acetylation, but two peaks were also added at 3000–2800 cm^−1^, expressing the C–H of polysaccharides compared with mycelia zinc polysaccharides, reported by Zhang et al. [11]. The structure of AMPS was further explained from different angles by HPLC and NMR spectrum analysis, which was beneficial to study the structure–activity relationship of polysaccharides. 

Moreover, the antioxidant activity of polysaccharides can be affected by the introduction of ion groups at appropriate levels [28]. The reducing power, as well as scavenging DPPH, hydroxyl, and superoxide radical activities, is widely used to investigate the antioxidant activity in vitro. Many literature pieces have reported that fungi polysaccharides have scavenging effects on these radicals, which can cause lipid peroxidation and further lead to various diseases [28,29,30]. Through this experiment, it was found that, when the substitution degree was 0.54 ± 0.04 in AMPS from *P. djamor*, the reducing power and the scavenging rates on the DPPH radicals, hydroxyl radicals, and superoxide anion radicals not only were increased in a dose-dependent way, but even had better antioxidant effects in vitro than mycelia zinc polysaccharides (MZPS), studied by Zhang et al. [12]. In this partial result, the antioxidant capacity in vitro of AMPS showed a more significant upward trend starting from the concentration of 500 mg/L to the last concentration gradients. The scavenging DPPH radical rate of polysaccharides even surpassed the dibutyl hydroxytoluene (BHT), an antioxidant reagent used as control group, beyond this concentration. 

There is increasing evidence that oxidative stress can obstruct the major antioxidant enzyme systems by reducing the activities of glutathione peroxidase, superoxide dismutase, and catalase [31]. GSH-Px is a selenoenzyme that can directly react with ROS to prevent the formation of hydroxyl radicals induced by H_2_O_2_. SOD can convert superoxide anion radicals into H_2_O_2_ to form a defense system of antioxidant enzymes against reactive oxygen species, thus playing an anti-aging role. CAT can catalyze the metabolism of hydrogen peroxide, decreasing activities of gaseous oxygen and water molecules. In addition, both MDA and LPO are considered to be the end products of lipid peroxidation and are markers of oxidative stress, leading to aging [32]. In this experiment, SOD, GSH-Px, and CAT activities were markedly elevated, while the contents of MDA and LPO were markedly depressed by administration of polysaccharides, indicating AMPS, especially at the dose of 400 mg/kg/d, has potential ability against d-gal-induced aging symptoms. The level of indicators in the blood can reflect the degree of damage of various tissues in the body, as well as the alleviating effect of AMPS on aging. ALT, CHE, TBIL, CREA, CK, AST, CHO, and GLU levels can be measured using a biochemical analyzer and were chosen as age-related indicators according to previous studies. The indicators above focused on the liver, kidney, and brain, with the aid of levels for lipids and glucose in serum as supplementary evidence for the first three tissues, thus guaranteeing the comprehensiveness and reliability of the data. ALT is released into the blood in large quantities when hepatocytes are necrotic in the acute phase, so it is an important indicator for the diagnosis of viral hepatitis [33]. The liver plays an important role in the metabolism of TBIL, including the three processes of uptake, binding, and excretion of unbound bilirubin in the blood by hepatocytes, and any obstacle in these processes can cause the accumulation of bilirubin in the blood [34]. When the corresponding liver cells are damaged, the permeability of the cell membrane increases and the AST in the cytoplasm is released into the blood, so the serum concentration of AST can be increased [35]. CREA is mainly excreted by glomerular filtration and its abnormal blood content indicates kidney damage in aging mice [36]. CK is mostly in the cytoplasm and mitochondria of myocardium, and is an important kinase that is directly related to intracellular energy movement, muscle contraction, and ATP regeneration, which are related to aging. Renal injury can be reflected to some extent with increasing content of CK [37]. Acetylcholinesterase has the strongest effect on the physiological concentration of acetyl choline (Ach) and higher specificity. The aging condition can be improved by restoring the content of acetylcholine in the cerebral cortex and reducing the activity of acetylcholinesterase [38]. Aging can lead to diabetes and other diseases. We explored the aging degree by detecting the glucose content in serum. In addition, CHO can be used as an indicator of blood lipids together with GLU to explore the efficacy of AMPS on d-gal-induced aging [39]. In the present study, each of the above eight indexes were decreased to a different degree compared with the MC group and some of them were even lower than that in the PC group at the dosage of 400 mg/kg/d of AMPS, indicating that AMPS has commendable anti-aging activity in vivo. Regarding serum immunology, NGF can regulate the growth and development of peripheral and central neurons and maintain the survival of them. TNF-α can kill or inhibit tumor cells, enhance the phagocytosis of neutrophils, and induce the synthesis of proteins in the acute phase of hepatocytes. Bcl-2 acts on near or far target cells with the help of lymphocytes, which is a very important way to achieve immune effects and immune regulation [40,41,42]. In addition, the polysaccharide dose groups showed good immunological activity, which also indicated that AMPS had anti-aging activity in vivo. Because the indexes can be measured by biochemical analyzer, in order to get more accurate data in immunology, the NGF, TNF-α, and Bcl-2 levels were measured by ELISA method mainly investigating brain injury, which could be used as a supplement to the CHE mentioned before. These three indicators just served as supplementary data for the value of the brain index measured, which was not obvious before. However, it could also reflect the extent of liver and kidney tissue damage caused by d-gal-induced aging to a certain extent. Compared with the model group, the histologic sections of liver, kidney, and brain under both doses of AMPS showed anesis in morphological features, indicating that the liver, kidney, and brain injuries were reduced. In other words, AMPS had a certain protective effect on them and was dose-dependent.

## 4. Materials and Methods

### 4.1. Strains and Culture Conditions

The fungus *P. djamor* strain was provided by Shandong Agricultural University and maintained by slants consisting of potato dextrose agar (PDA). The *P. djamor* in Petri dishes were inoculated into 1000 mL suction flasks containing PDA liquid medium (500 mL) and incubated on a rotary shaker at 130 rpm for 7 d at 25 °C. Further, mycelia were acquired from fermentation cylinder by filtration.

### 4.2. Preparation of AMPS

The mycelia polysaccharides of (MPS) were prepared according to the method of Liu et al. [43], with slight modifications. The mycelia powder of *P. djamor* was mixed with proper distilled water for 4 h at 90 °C, then centrifuged at 3000 rpm for 10 min. The supernatant was concentrated, mixed three times with ethanol (95%, *v*/*v*), and then kept at 4 °C overnight. After centrifugation (3000 rpm, 10 min), the precipitate was collected and dried at 55 °C. The crude polysaccharides were deproteinized by employing the Sevage method [44] and lyophilized to obtain MPS. 

The acetylated polysaccharides were obtained ulteriorly according to Jia et al. [45] with minor modifications. The reaction time, reaction temperature, and the amount of acetic anhydride were selected for the single-factor tests and response surface optimization experiment using the degree of substitution (DS) for response values. 

The powered MPS (400 mg) was dissolved in distilled water (pH 9.0), adjusted by sodium hydroxide solution (0.1 mol/L), and the acetylation procedure was processed according to the optimized conditions. The acetylated solution was neutralized with hydrochloric acid (0.1 mol/L) after this reaction. The AMPS was obtained after centrifugation (3000 rpm, 10 min), dialysis (Molecular weight cut off, MWCO = 3500, 4 °C, 3 d), concentration (80 °C), and lyophilization (3 d) (Beili, Beijing, China and SCIENTZ, Ningbo, China) and was considered to be AMPS. 

### 4.3. The DS Assay of AMPS

#### 4.3.1. Standard Curve Plotting of Acetyl Content

The DS of acetylation substitution was measured as previously reported [46], with a little adjustment. β-d-penta-acetyl glucopyranose (0.6978 g, Mw = 390.34 u) was dissolved into 20 mL of ethyl alcohol adequately by 60 °C water-bath heating, then the solution was kept at a constant volume and shaken up in a 100 mL volumetric flask after cooling down to room temperature. Volumes of 2 mL, 4 mL, 6 mL, 8 mL, 10 mL, and 12 mL were removed from the aforementioned β-d-penta-acetyl glucopyranose stock solution, respectively, and the 50 mL volumetric flask was used to keep the volume constant. The β-d-penta-acetyl glucopyranose standard solution was obtained after further dilution 10 times. The absorbance at 500 nm was acquired based on the series of concentration of acetyl.

#### 4.3.2. Calculation of AMPS Acetyl Substitution Degree

A brown volumetric flask containing 10 mL sample solution (1 mg/mL), 5 mL hydroxylamine hydrochloride (0.1 mol/L), and 5 mL sodium hydroxide solution (0.15 mol/L) was placed for 20 min. The above solution was neutralized and ferric chloride (0.37 mol/L, 10 mL) was added after 20 min. The absorbance at 500 nm was measured after constant volume by deionized water. The acetylation substitution degree refers to the number of hydroxyl groups substituted by an acetyl group in average glucose loss per unit. The DS is expressed as
w_2_ (%) = W_2/_W_1_ × 100,(1)
DS = 162 × w_2_/(4300 − 42 × w_2_),(2)
where w_2_ is the mass fraction of the acetyl (%), W_1_ is the quality of AMPS (mg), and W_2_ is the acetyl quality of AMPS (mg).

### 4.4. FT-IR, HPLC, and NMR Analysis

The AMPS (100 mg) was mixed with KBr powder (100–200 mg) and then pressed into pellets for infrared spectral analysis within a range from 4000 to 400 cm^−1^. The FT-IR spectrum was measured using a 6700 Nicolet Fourier transform-infrared spectrophotometer (Thermo Co., Madison, WI, USA).

The molecular weights were determined by HPLC, which was performed with an HPLC system (Aglient-1260 Infinity, Agilent Technologies, CA, USA) equipped with SB-802.5HQ, SB-804HQ column, an eight-angle laser light scatterometer (BI-MWA, Brookhaven instruments, Inc, USA), and a refractive index detector. The injection volume was 100 μL. The NaCl and Na_2_HPO_4_ aqueous solution was used as the mobile phase at a flow rate of 0.8 mL/min and the column temperature was maintained at 25 °C. A series of standard dextrans were used to make the calibration curve. Molecular weight was analyzed by Agilent GPC software (Agilent 1260 Infinity Multi-Detector GPC/SEC System, A.02.01).

^1^H and ^13^C-NMR measurements were conducted using a Bruker AV-300 spectrometer operating at 700 MHz and 25 °C and the sample was dissolved in deuterated water (D_2_O).

The monosaccharide composition of AMPS was determined by an HPLC system (Ultimate 3000, Agilent Technologies, CA, USA) equipped with Xtimate C_18_ column (4.6 mm × 200 mm × 5 um). The injection volume was 20 μL. The 0.05 mol/L potassium dihydrogen phosphate–acetonitrile solution was used as the mobile phase at a flow rate of 1.0 mL/min, the column temperature was maintained at 30 °C, and the test wavelength was 250 nm. The mannose, ribose, rhamnose, glucuronic acid, galacturonic acid, glucosamine, glucose, galactose, galactose, xylose, arabinose, and fucose were critically weighed and diluted to 50 μg mixed control solution in 1 mL each. After derivation of the sample solution and the mixed control solution, the monosaccharide composition of AMPS was explored.

### 4.5. Antioxidant Activity In Vitro

#### 4.5.1. Reducing Power Assay

The reaction mixtures consisted of 1 mL AMPS (100–1400 mg/L), 2.5 mL phosphate buffer solution (pH 6.6, 0.2 mol/L), and 1 mL potassium ferricyanide (1%, *w*/*v*). After incubating at 50 °C for 20 min, 2 mL trichloroacetic acid (10%, *w*/*v*) and 1.2 mL ferric trichloride (1%, *w*/*v*) were added to the mixture. Thenm the absorbance was measured at 700 nm under the condition of zeroing by deionized water. The EC_50_ value (mg/L) was defined as the effective concentration of the sample at which reducing power reached half of the maximum.

#### 4.5.2. Scavenging DPPH Radical Assay

The scavenging DPPH radical activity was determined according to Sun and Ho [47]. The reaction mixture contained DPPH (2 mL, 0.2 mmol/L) and the sample (2 mL, 100–1400 mg/L). After shaking vigorously and incubating in the dark for 30 min, the absorbance was measured at 517 nm. A mixture of sample (2 mL) and absolute ethyl alcohol (2 mL) as blank group, deionized water (2 mL) and DPPH (2 mL) as a control group and the compound of deionized water (2 mL) and absolute ethyl alcohol (2 mL) was used for zero setting in this trial. The scavenging DPPH radical rate was expressed as
Scavenging ability (%) = [1 − (Ai − Aj)/Ac],(3)
where Ai is the absorbance of the tested sample, Aj is the absorbance of the blank, and Ac is the absorbance of the control. The IC_50_ value (mg/L) was defined as the effective concentration of the sample at which DPPH radicals were inhibited by 50%.

#### 4.5.3. Scavenging Hydroxyl Radical Assay 

The scavenging hydroxyl radical activity was determined according to Smirnoff and Cumbes [48], with some modifications. The H_2_O_2_ (1 mL, 8.8 mmol/L) was added to the reaction system containing FeSO_4_ (1 mL, 9 mmol/L), ortho-hydroxybenzoic acid (1 mL, 9 mmol/L), and the sample (1 mL, 100–1400 mg/L) at 37 °C for 30 min. The absorbance was measured at 510 nm after centrifugation (1200 rpm, 6 min). An equal amount of deionized water replaced the sample in this system as the control group and zeroed with deionized water. The scavenging hydroxyl radical activity was expressed as
Scavenging activity (%) = [1 − (Aj − Ai)/Aj],(4)
where Ai is the absorbance of the tested sample and Aj is the absorbance of the blank. The IC_50_ value (mg/L) was defined as the effective concentration of the sample at which hydroxyl radicals were inhibited by 50%. 

#### 4.5.4. Scavenging Superoxide Radical Assay

The 11 test tubes were taken and numbered and 0.5 mL phosphate buffer solution was added (pH 7.8, 0.2 mol/L), along with 0.5 mL riboflavin (10 mmol/L), 0.25 mL methionine (13 mmol/L), and 0.25 mL nitroblue tetrazolium (15 mmol/L), in order. Then, different concentrations of polysaccharide diluent (100–1400 mg/L) and deionized water were accreted into the 11 systems respectively (1 mL), reacting under 25 °C for 20 min after blending. Absorbance values of the supernatant from the 11 systems at 560 nm were measured by using deionized water as a reference. The scavenging superoxide radical rate was evaluated according to the formula
Scavenging activity (%) = [1 − (Aj − Ai)/Aj],(5)
where Ai is the absorbance of the tested sample and Aj is the absorbance of the blank. The IC_50_ value (mg/L) was defined as the effective concentration of the sample at which superoxide radicals were inhibited by 50%. 

### 4.6. Oral Chronic Toxicity Test

Twelve male Kunming strain mice were collected for the acute toxicity study. The mice were divided into two groups with six in each group and the mice were gavaged with AMPS at dosages of 300 and 900 mg/kg/d, respectively. The mice were observed continuously for toxic symptoms, mortality, and behavioral changes during the whole feeding period for 2 weeks.

### 4.7. Anti-Aging Effect In Vivo

#### 4.7.1. Animal Experiments

Fifty Kunming mice (20 ± 2 g) were purchased from Taibang Biological Products Ltd. Co. (Taian, China) and the animal experiments were approved through the institutional animal care and use committee of Shandong Agricultural University in accordance with the Animals (Scientific Procedures) Act of 1986 (amended 2013). The mice were acclimatized for 7 days under controlled conditions (20–25 °C, lights on 12 h daily) with diet and water ad libitum. The mice of the control groups were randomly allocated into normal (NC), positive (PC), and model (MC) groups (ten mice in each group). The test group was further randomly and equally divided into the high dosage group (H-AMPS, 400 mg/kg/d) and low dosage group (l-AMPS, 200 mg/kg/d). The normal group was administered 0.2 mL normal saline and the other groups were administered 0.2 mL d-gal (300 mg/kg/d) through intraperitoneal injection. After 8 h, the normal and model groups were treated with 0.2 mL normal saline, the PC group was treated with 0.2 mL ascorbic acid (Vc, 300 mg/kg/d) through gastric gavage, and the test groups were gavaged with 0.2 mL of different concentration polysaccharides. After six weeks, the mice were fasted overnight and sacrificed through exsanguination under diethyl ether anesthesia. 

#### 4.7.2. Biochemical Analysis 

The livers, kidneys, and brains were rapidly removed, weighed, and immediately homogenized (1:9, *w*/*v*) in phosphate buffer solution (0.2 M, pH 7.4, 4 °C). After centrifugation (5000 rpm, 4 °C) for 20 min, the supernatants were collected for further biochemical analysis. The activities of GSH-Px, SOD, and CAT, as well as the contents of MDA and LPO in the liver, kidney, and brain, were assayed using commercially available diagnostic kits (Nanjing Jiancheng Bioengineering Institute, Nanjing, China).

The blood samples obtained from the retrobulbar veins were centrifuged at 14,000 rpm at 4 °C for 10 min to gain the serum. The activities of ALT, CHE, CK, and AST and the levels of TBIL, CREA, CHO, and GLU were analyzed using an automatic biochemical analyzer (ACE, USA). Moreover, the NGF, TNF-α, and Bcl-2 levels in the serum were evaluated by commercial kits, according to independent instructions of the ELISA method.

#### 4.7.3. Histopathological Analysis

According to the method of Zhu et al. [49], the livers, kidneys, and brains were removed and rapidly immersed into formalin solution (10%, *v*/*v*, pH 7.4) with the purpose of buffering and fixation for 7 days. Then the tissues of each group were divided into small tea bags with labels. After three days of running water flushing, the tissues were rapidly put into ethanol (50%), further dehydrated by gradient ethanol solution, and then administered transparent processing with xylene. After immersion in paraffin (3 times successively for 2 h each time) and embedding, tissues were cut into 5 μm slices. The glass slides were accompanied by paraffin section into 60 °C oven for 2 days. Xylene and gradient ethanol were used for deparaffinating and hydration, respectively. The samples were observed under the microscope and photographed at 400× magnification to evaluate the morphological changes after hematoxylin–eosin (H and E) staining, distilled water washing, gradient ethanol, and xylene treatment, in sequence. 

### 4.8. Statistical Analysis

All experiments were performed in triplicate and the results are presented as the means ± standard deviations (SD). The results were analyzed using one-way analysis of variance (ANOVA) with the IBM SPSS Statistical software package program. *p* < 0.05 was considered statistically significant.

## 5. Conclusions

This experiment proved that the AMPS from *P. djamor* had good antioxidant activity in vitro, anti-aging activity in vivo, and protective effects on the liver, kidney, and brain. It was verified that acetylation was a suitable modification method for the mycelia polysaccharides from *P. djamor*, indicating that AMPS can be used as a functional food and nutritional medicine for the treatment of aging and age-related diseases.

## Figures and Tables

**Figure 1 molecules-24-02698-f001:**
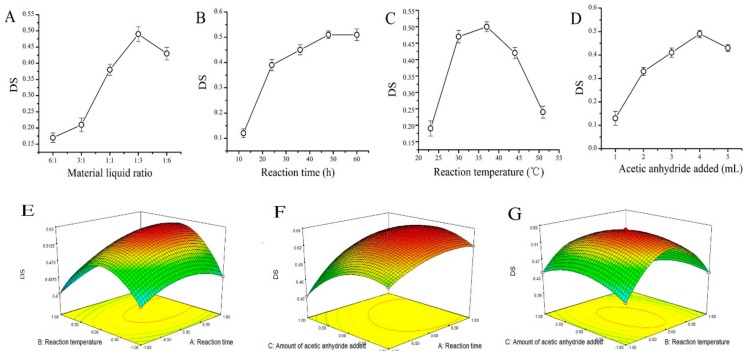
Results of single–factor and response surface optimization experiment. Influences of (**A**) material ratio, (**B**) reaction time, (**C**) reaction temperature, and (**D**) the amount of acetic anhydride added to DS in single-factor test. Influences of (**E**) reaction time and reaction temperature, (**F**) reaction time and the amount of acetic anhydride added, and (**G**) reaction temperature and the amount of acetic anhydride added to DS in the response surface optimization experiment. Factors that were not included in the axes were fixed at their respective optimum levels. The values were reported as the means ± SD (*n* = 10 for each group). Means with the same letter had no significant difference (*p* < 0.05). DS: substitution degree.

**Figure 2 molecules-24-02698-f002:**
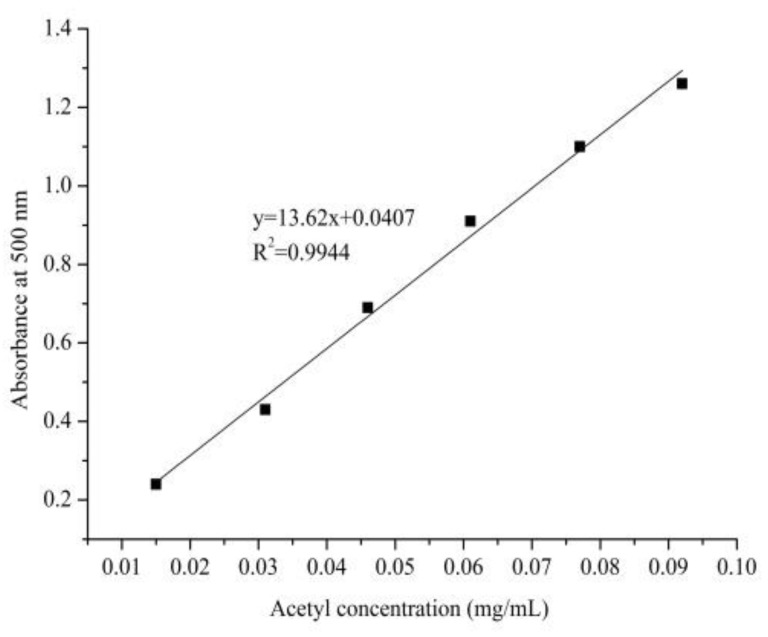
Standard curve of acetyl content.

**Figure 3 molecules-24-02698-f003:**
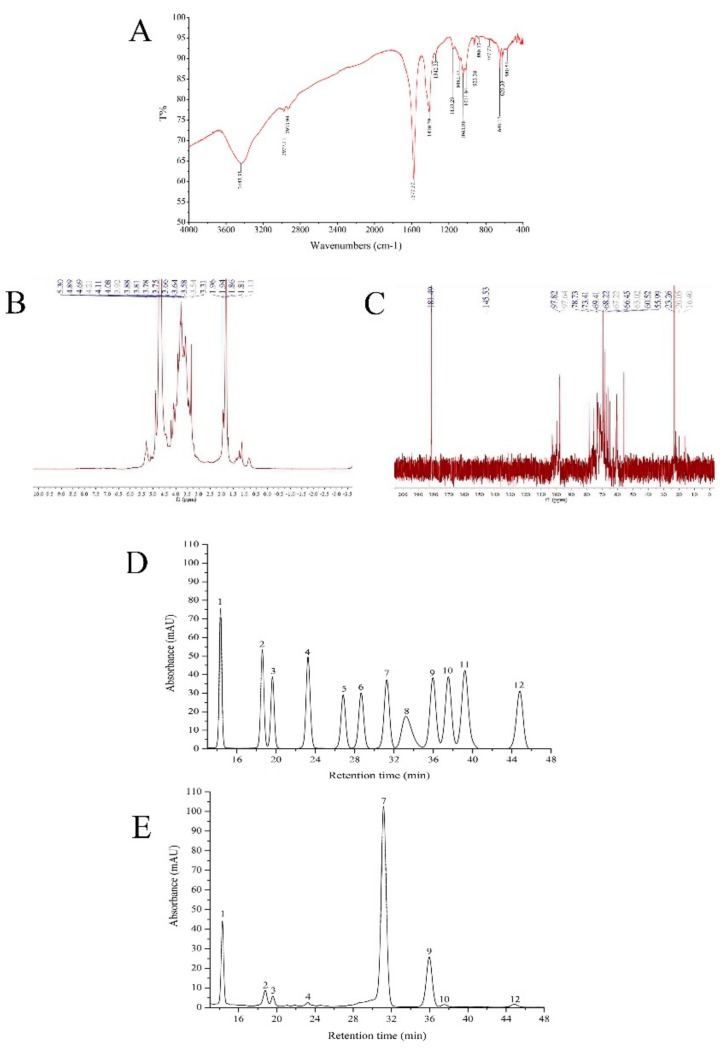
Physicochemical analysis of acetylated mycelia polysaccharides (AMPS). (**A**) FT-IR spectra over the range of 400–4000 cm^−1^, (**B**) NMR analysis of ^1^H spectra and (**C**) ^13^C spectra, and (**D**) gas chromatography of 12 standard monosaccharides. Peaks: (1) Mannose, (2) ribose, (3) rhamnose, (4) glucuronic acid, (5) galacturonic acid, (6) glucosamine, (7) glucose, (8) galactose, (9) galactose, (10) xylose, (11) arabinose, and (12) fucose. (**E**) Gas chromatography of AMPS.

**Figure 4 molecules-24-02698-f004:**
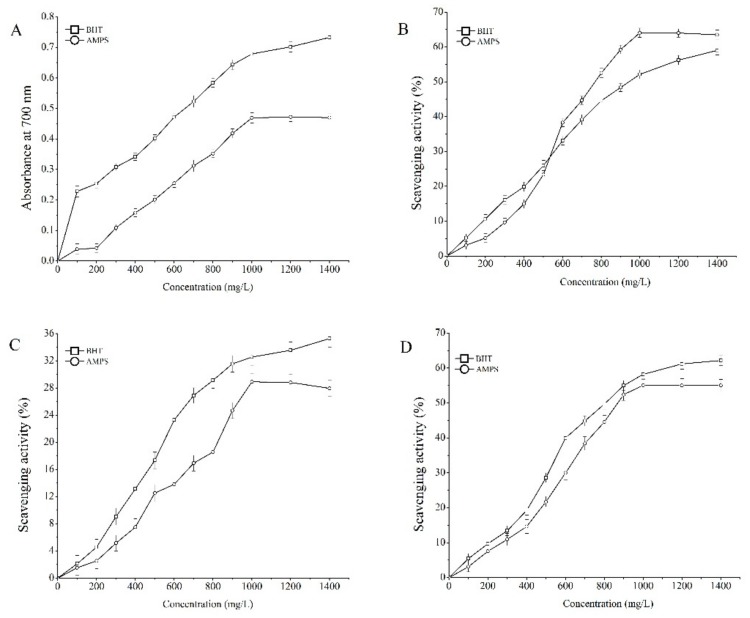
Antioxidant activities of AMPS in vitro. (**A**) Reducing power, (**B**) scavenging ability on 1,1-diphenyl-2-picrylhydrazyl (DPPH) radicals, (**C**) scavenging ability on hydroxyl radicals, and (**D**) scavenging ability on superoxide anion radicals. The values were reported as means ± SD (*n* = 10 for each group). Means with the same letter had no significant difference (*p* < 0.05).

**Figure 5 molecules-24-02698-f005:**
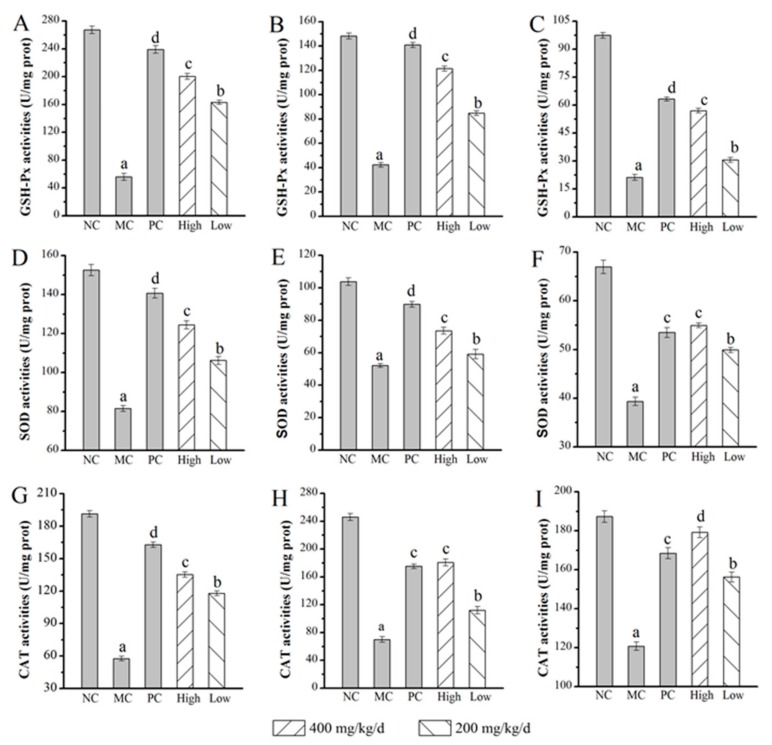
Effects of AMPS on the activities in liver of GSH-Px (**A**), SOD (**D**), and CAT (**G**), in kidney of GSH-Px (**B**), SOD (**E**), and CAT (**H**), and in brain of GSH-Px (**C**), SOD (**F**), and CAT (**I**), respectively. The values were reported as means ± SD (*n* = 10 for each group). Means with the same letter had no significant difference (*p* < 0.05). SOD: superoxide dismutase, GSH-Px: glutathione peroxidase, and CAT: catalase.

**Figure 6 molecules-24-02698-f006:**
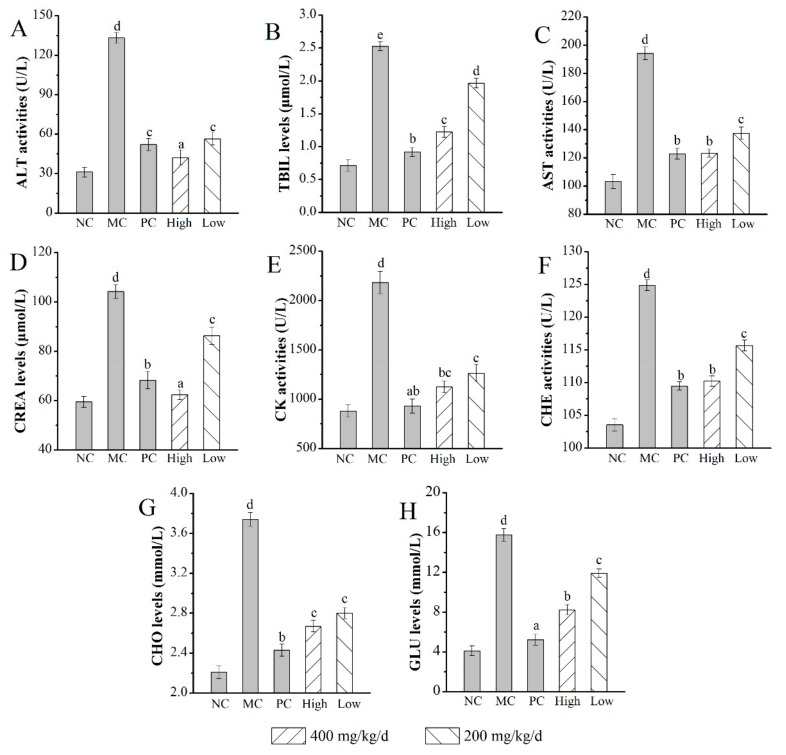
Effects of AMPS on (**A**) ALT activities, (**B**) TBIL levels, (**C**) AST activities, (**D**) CREA levels, (**E**) CK activities, (**F**) CHE activities, (**G**) CHO levels, and (**H**) GLU levels in d-gal-induced aging mice. The values were reported as means ± SD (*n* = 10 for each group). Means with the same letter had no significant difference (*p* < 0.05). ALT: alanine transaminase, CHE: cholinesterase, CK: creatine kinase, AST: aspartate aminotransferase, TBIL: total bilirubin, CREA: creatinine, CHO: total cholesterol, and GLU: glucose.

**Figure 7 molecules-24-02698-f007:**
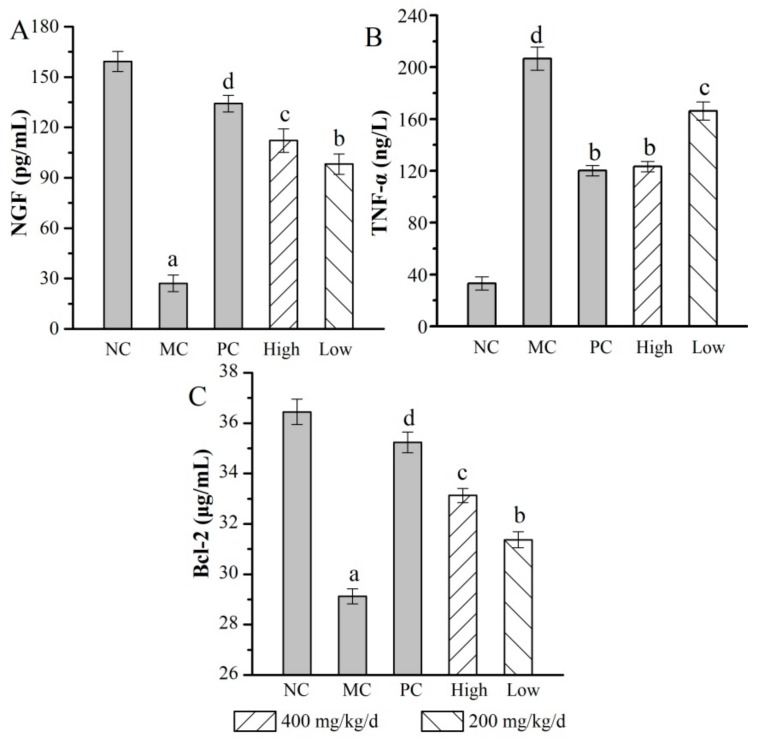
Effects of AMPS on the levels of (**A**) NGF, (**B**) TNF-α, and (**C**) Bcl-2 in d-gal-induced aging mice. The values were reported as means ± SD (*n* = 10 for each group). Means with the same letter had no significant difference (*p* < 0.05). NGF: nerve growth factor, TNF-α: tumor necrosis factor-α, and Bcl-2: B-lymphoma factor-2.

**Figure 8 molecules-24-02698-f008:**
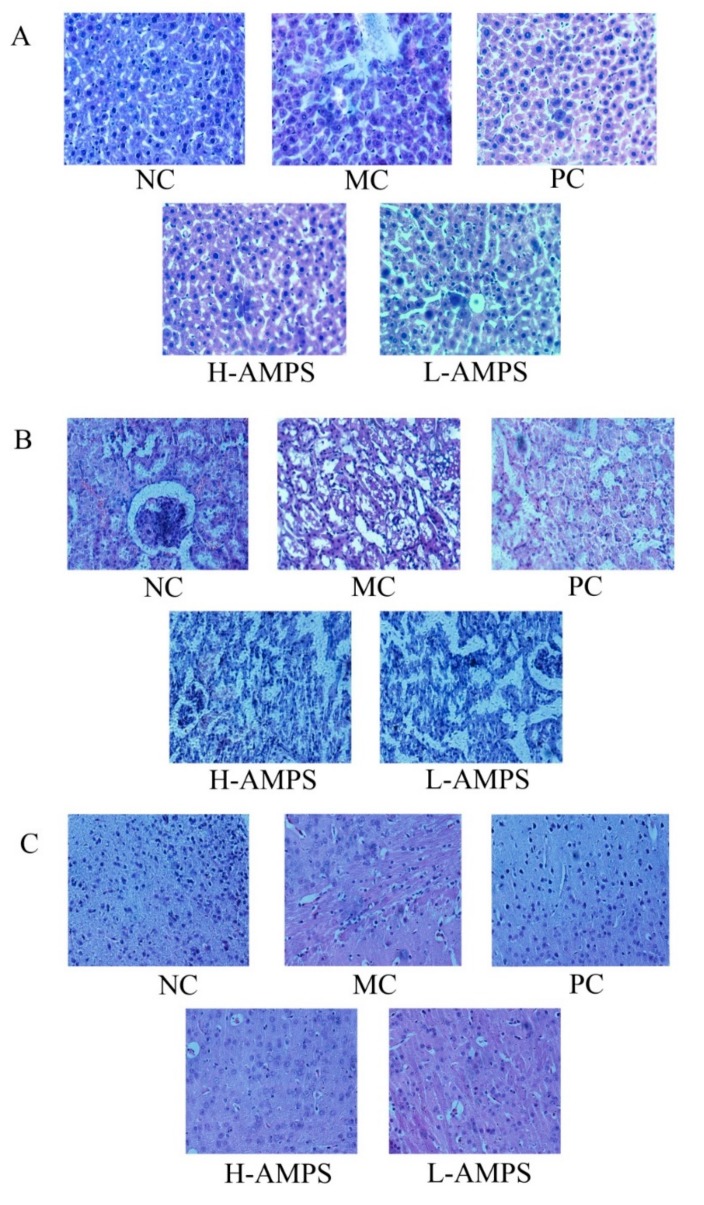
Effects of AMPS on organ damage of (**A**) liver, (**B**) kidney, and (**C**) brain in d-gal-induced aging mice at the dosage of 400 mg/kg/d (H-AMPS), showing slight architectural damage, and 200 mg/kg/d (l-AMPS), showing mild architectural damage (hematoxylin–eosin staining, magnification 400×).

**Table 1 molecules-24-02698-t001:** Effects of AMPS on body weight, liver, and kidney indexes in d-gal-induced aging mice.

Groups	Body Weight (g)		Liver Index (g/100g)	Kidney Index (g/100g)	Brain Index (g/100g)
	Initial	Final			
NC	24.72 ± 1.21	41.44 ± 1.28	7.54 ± 0.21	3.78 ± 0.05	1.05 ± 0.05
MC	25.11 ± 0.97	32.89 ± 1.44 ^a^	13.38 ± 0.19 ^d^	6.33 ± 0.09 ^e^	3.79 ± 0.12
PC	24.98 ± 1.12	39.63 ± 1.57 ^bc^	8.26 ± 0.14 ^b^	3.94 ± 0.04 ^b^	1.12 ± 0.06
AMPS					
400 mg/kg/d	25.02 ± 1.45	38.91 ± 1.31 ^bc^	8.96 ± 0.22 ^b^	4.25 ± 0.04 ^c^	1.56 ± 0.04
200 mg/kg/d	24.35 ± 1.34	37.04 ± 1.19 ^b^	9.21 ± 0.18 ^c^	4.98 ± 0.06 ^d^	2.11 ± 0.09

The values were reported as means ± SD (*n* = 10 for each group). Means with the same letter had no significant difference (*p* < 0.05). NC: normal control, MC: model control, and PC: positive control.

**Table 2 molecules-24-02698-t002:** Effects of AMPS on the MDA and LPO contents of the liver, kidney, and brain.

Groups	Groups MDA Contents (μmol/mg prot)			LPO Contents (nmol/mg prot)		
	Liver	Kidney	Brain	Liver	Kidney	Brain
NC	3.13 ± 0.19	4.92 ± 0.21	2.02 ± 0.09	4.21 ± 0.19	3.80 ± 0.11	1.84 ± 0.08
MC	13.21 ± 0.45 ^d^	15.73 ± 0.51 ^d^	15.09 ± 0.58 ^e^	17.23 ± 0.59 ^e^	16.87 ± 0.51 ^d^	7.44 ± 0.39 ^d^
PC	3.61 ± 0.12 ^a^	6.24 ± 0.24 ^b^	8.09 ± 0.31 ^b^	5.92 ± 0.49 ^b^	4.71 ± 0.24 ^a^	2.24 ± 0.12 ^a^
AMPS						
400 mg/kg/d	5.70 ± 0.19 ^b^	6.12 ± 0.22 ^b^	10.05 ± 0.38 ^c^	9.89 ± 0.32 ^c^	8.81 ± 0.29 ^b^	4.28 ± 0.23 ^b^
200 mg/kg/d	9.35 ± 0.34 ^c^	9.51 ± 0.38 ^c^	12.03 ± 0.49 ^d^	12.31 ± 0.39 ^d^	11.95 ± 0.42 ^c^	5.67 ± 0.17 ^c^

The values were reported as means ± SD (*n* = 10 for each group). Means with the same letter had no significant difference (*p* < 0.05). MDA: malondialdehyde, LPO: lipid peroxide and prot: protein, respectively.

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
