# Peer review of "The Antioxidant and Anti-Aging Effects of Acetylated Mycelia Polysaccharides from Pleurotus djamor"

_molecules, 2019, doi:10.3390/molecules24152698_

Reviewer 1 Report

This manuscript reports a new preparation of P. djamour and its anti-aging properties using in vivo and in vitro experiments. Thus, biochemical markers demonstrated that AMS treatment may have the ability to enhance the immune function working as a bioactive compound. The manuscript is well written with methods and results properly discussed. My only overall suggestion is that we can not find the reason why authors defined dose response effect just testing two concentrations. It is missing IC50 data. Moreover, it would be very interesting to correlate this results with previous findings from the same group analyzing antioxidant properties. 

Author Response

Thanks for your kind suggestion. The IC50 refers to the half-dose that indicates that a drug or substance inhibits certain biological processes. The EC50 refers to concentration for 50% of maximal effect. The relevant content has been supplemented in the manuscript. At the concentration of 1000 mg/L, the absorbance at 700 nm reached 0.47 ± 0.02 and the EC50 level was 620 mg/L. At the concentration of 1000 mg/L, the scavenging DPPH radical effect was 64.09 ± 1.34% and the IC50 level was 580 mg/L. The scavenging hydroxyl radical value reached 28.91 ± 1.24% under the concentration of 1000 mg/L and the IC50 level was 715 mg/L. The scavenging rate reached 55.05 ± 1.96% at 1000 mg/L and the IC50 level was 585 mg/L.

Reviewer 2 Report

The work presented for the review is interesting and well written, however, it needs to be supplemented and improved in point 2.3. Firstly, there is no positive control. Second - the results are presented as a percentage of activity. Percentage is not an SI unit. The results should be presented as an equivalent of e.g. Trolox or as EC50. The authors state that "AMPS had a potential scavenging result with concentration-dependent manners". This is not entirely true, because such a dependence was found only in the studied range, and the maximal activity did not exceed 60%. Have tests been conducted for concentrations greater than 1000 mg / L?

Author Response

Thanks for your appreciation and affirmation of our manuscript. The EC50 refers to concentration for 50% of maximal effect. The IC50 refers to the half-dose that indicates that a drug or substance inhibits certain biological processes. The antioxidant properties in vitro of polysaccharides whose concentration were higher than 1000 mg/L was also detected, and there was not obvious upward trend for reducing power and scavenging radicals capacities. The figure 4 was improved in the manuscript.

Reviewer 3 Report

1.     In abstract and whole MS the SD value is not need.

2.      Abstract: First sentence: This paper.. it is not good beginning of abstract, please to correct.

3.     Abstract, 21 line: food for treatment of aging.. should be in prevention of aging.

4.     Keywords should be another than used in title of MS.

5.     P. djamor, in vitro, in vivo should be in italic in whole MS.

6.     181 line please to correct this sentence: normal…..unnormal

7.     378 line: Materials and methods: is lack of data about mushrooms identification, abbreviation of authors names and families of selected for study species.

379 line; The fungus strain P. djamor, should be: P. djamor strain;

The same section please to correct for better readable.

8. All figures should be corrected for better readable.

Author Response

1. In abstract and whole MS the SD value is not need.

For the rigor of the experiment and the requirement of statistical analysis, all experiments were performed in triplicate, and the results are presented as the means ± standard deviation (SD). The results were analyzed using one-way analysis of variance (ANOVA) with the IBM SPSS Statistical software package programme. P < 0.05 was considered statistically significant. Hence, the SD values was academically necessary for the Ms..

2. Abstract: First sentence: This paper. it is not good beginning of abstract, please to correct.

Thanks for your useful suggestion. “This paper” has been corrected to “The present work”.

3. Abstract, 21 line: food for treatment of aging.. should be in prevention of aging.

Thanks for your valuable comments. The “treatment” has been changed to the “prevention” for emphasize the effect of polysaccharide on preventing aging in advance.

4. Keywords should be another than used in title of MS.

Thanks for your professional advice. The “Injury of liver; kidney and brain” has been removed from the keywords, and the “Antioxidation” and “Anti-aging” was also corrected to “Antioxidant effects” and “Anti-aging effects”.

5. P. djamor, in vitro, in vivo should be in italic in whole MS.

Thanks a lot for your meaningful comments. The three word groups have been transformed into “P. djamor” “in vitro” and “in vivo”, respectively.

6. 181 line please to correct this sentence: normal…..unnormal.

Thanks a lot for your technical comments. This sentence has been corrected as “When compared to the normal control group, mice in model control group showed some abnormal behavior characteristics, such as slow reaction, reduced feeding, and sparse hair and the death ratio was higher, suggesting the success of aging model construction in the behavioral aspects of aging mice.” in the manuscript.

7. 378 line: Materials and methods: is lack of data about mushrooms identification, abbreviation of authors names and families of selected for study species. 379 line; The fungus strain P. djamor, should be: P. djamor strain; The same section please to correct for better readable.

Thanks for your kind comments. The fungus P. djamor strain used in present work was provided from Shandong Agricultural University, and the original strain was purchased from China General Microbiological Culture Collection Center with the CGMCC ID of 5.673.

8. All figures should be corrected for better readable.

Thanks for your comments. The resolution of the figures in the manuscript has been improved for better readable.

Round  2

Reviewer 2 Report

OK